# A Rich Array of Prostate Cancer Molecular Biomarkers: Opportunities and Challenges

**DOI:** 10.3390/ijms20081813

**Published:** 2019-04-12

**Authors:** Indu Kohaar, Gyorgy Petrovics, Shiv Srivastava

**Affiliations:** 1Henry Jackson Foundation for the Advancement of Military Medicine (HJF), Bethesda, MD 20817, USA; 2Center for Prostate Disease Research, Department of Surgery, Uniformed Services University of the Health Sciences and the Walter Reed National Military Medical Center, Bethesda, MD 20814, USA; shsr629@gmail.com

**Keywords:** prostate cancer, molecular biomarkers, diagnosis, prognosis

## Abstract

Prostate cancer is the most prevalent non-skin cancer in men and is the leading cause of cancer-related death. Early detection of prostate cancer is largely determined by a widely used prostate specific antigen (PSA) blood test and biopsy is performed for definitive diagnosis. Prostate cancer is asymptomatic in the early stage of the disease, comprises of diverse clinico-pathologic and progression features, and is characterized by a large subset of the indolent cancer type. Therefore, it is critical to develop an individualized approach for early detection, disease stratification (indolent vs. aggressive), and prediction of treatment response for prostate cancer. There has been remarkable progress in prostate cancer biomarker discovery, largely through advancements in genomic technologies. A rich array of prostate cancer diagnostic and prognostic tests has emerged for serum (4K, phi), urine (Progensa, *T2-ERG*, ExoDx, SelectMDx), and tumor tissue (ConfirmMDx, Prolaris, Oncoytype DX, Decipher). The development of these assays has created new opportunities for improving prostate cancer diagnosis, prognosis, and treatment decisions. While opening exciting opportunities, these developments also pose unique challenges in terms of selecting and incorporating these assays into the continuum of prostate cancer patient care.

## 1. Introduction

Prostate cancer is a major cause of morbidity and mortality worldwide. It is the second most frequent cancer and the fifth leading cause of cancer death in men [1]. In the United States prostate cancer is the most prevalent male cancer with an estimated 174,650 new cases and ranks second in relation to cancer related cancer deaths, with 31,620 deaths in 2019 [2]. Notably, prostate cancer incidence is the highest among men of African ancestry, followed by European and Asian men [3,4,5,6]. The rationale for selecting a biomarker in the oncology field includes its ability to predict early detection, therapeutic response, and staging, in addition to its ability to reduce over-diagnosis, distinguish lesion types (to identify sub-classes of prostate cancer), select patients for different treatment options, increase life expectancy, and provide a better quality of life for the patient. A biomarker is most often classified by its application. Screening or early detection biomarkers may predict the disease during a man’s asymptomatic state of cancer progression. Diagnostic biomarkers can predict cancer in patients suspected of having a disease, while prognostic biomarkers predict the course of disease progression. Predictive biomarkers can predict risk of disease onset/progression or response to a given therapy in a subset of the patient population. Lastly, surrogate biomarkers can be used to substitute for a clinical endpoint and/or to measure clinical benefit, harm, or lack of benefit or harm to the patient. Many prostate cancer molecular biomarkers have been identified; however, only a few are Food and Drug Administration (FDA) approved (PSA in 1994, phi in 2012, and *PCA3* in 2012) (Table 1). 

An ideal diagnostic biomarker has high specificity (ability of the test to correctly identify those without the disease; true negative rate), high sensitivity (ability of a test to correctly identify those with the disease; true positive rate), simplistic ease-of-use, reproducibility, clear read-outs for clinicians, cost-effectiveness, and quantifiable measures from an easy-to-obtain biological fluid or specimen. The present review is based on the current state of knowledge of available diagnostic and prognostic molecular markers in prostate cancer, as well as those under development, and discusses their utility in the clinic. 

## 2. Blood-Based Biomarkers

### 2.1. Prostate Specific Antigen (PSA)

Prostate specific antigen (PSA) is a 33 kDa serine protease encoded by the Kallikrein 3 (*KLK3*) gene on chromosome 19q 13.3–13.4. It is produced and secreted by the ductal and acinar epithelium of the prostate. PSA discovery has been marked as one of the landmark discoveries of the 20th century. Dr. T. Ming Chu and colleagues were the first group to purify and characterize PSA and to show that it was prostate-specific [7]. The PSA was initially identified and characterized as a marker of human semen [8,9]. The qualitative and quantitative differences of serum PSA may differentiate between carcinoma and benign inflammatory disease of the prostate. For example, 65% to 95% of total PSA (tPSA) is bound to protease inhibitors, such as α-1-antichymotrypsin (PSA-ACT); however, the molecular form of PSA that is not found in a protein complex is referred to as free PSA (fPSA). A low ratio of fPSA/tPSA has been identified as a characteristic of prostate cancer, but the differences in the stabilities of the molecular forms of PSA have cautioned the use of PSA as a primary screening method in the clinic [10]. The PSA test was originally approved by the FDA in 1986 to monitor the progression of prostate cancer in men who had already been diagnosed with the disease. In 1994, the FDA approved the serum PSA test, in combination with a digital rectal exam (DRE), for diagnostic screening of prostate cancer in the clinic [11]. Importantly, PSA screening has significantly contributed to early detection and treatment, resulting in a decline in prostate cancer mortality over the last three decades [12,13]. Interestingly, in the early days of PSA testing, African American (AA) men with newly diagnosed prostate cancer had higher serum PSA levels at initial diagnosis than Caucasian American (CA) men, which corresponded to a larger tumor volume observed in AA men treated in an equal access health care system [14] . This stage-for-stage tumor volume disparity even in an equal-access health care environment may relate to screening behavior and/or biological differences of prostate cancer in black men [15,16,17,18]. Also, we have very limited data on race specific biomarkers in prostate cancer as most of the biomarker studies are primarily based on CA men. Therefore, it is important to evaluate the utility of available markers together with an exploration of new biomarkers in the context of race. Although a serum PSA threshold of 4 ng/mL is still used as a cut-off in patients with an abnormal DRE, the observation of rising PSA values over time, even at levels below 4 ng/mL, is found to be a more significant diagnostic parameter. Even though individuals with persistently elevated PSA levels eventually undergo biopsy, it is an imprecise marker for cancer risk, because higher serum PSA levels do not always correspond to the presence of cancer [19]. Non-cancerous pathologic conditions, such as prostatitis and benign prostatic hyperplasia (BPH), could also trigger a serum PSA rise, leading to unnecessary biopsies [20,21].

In 2012, the U.S. Preventive Services Task Force (USPSTF) gave PSA-based testing a “D” grade for prostate cancer screening, regardless of age within the US population [22,23]. This recommendation did not include the use of the PSA test for surveillance after diagnosis or treatment of prostate cancer, where the test has much higher accuracy. Instead, the use of the PSA test for this indication was concluded outside of the scope of the USPSTF [24]. Unnecessary biopsies, performed because of abnormal PSA levels, led to a myriad of complications in patients, among which erectile dysfunction and urinary incontinence are the most common [25]. Since then, there have been continuums of debates over the harms and benefits of PSA-based screening for prostate cancer. As of May 2018, the revised “C” grade recommendation of the USPSTF for PSA screening states, “for men aged between 55–69 years the decision to undergo periodic PSA based screening for prostate cancer should be an individualized one. Before deciding whether to be screened, men should have an opportunity to discuss the potential benefits and harms of screening with their clinician and to incorporate their values and preferences in the decision”; however, USPSTF still recommends against PSA-based screening for prostate cancer in men aged 70 years and older [26].

### 2.2. Prostate Health Index (phi)

The Prostate Health Index (phi; Beckman Coulter, Brea, CA, USA) is a blood-based test that analyzes levels of fPSA, tPSA, and [–2]proPSA (p2PSA) to predict risk of Gleason ≥7 disease at biopsy [27]. The phi assay was approved by the FDA in 2012 to distinguish prostate cancer from other benign diseases in men aged 50 years, having non-suspicious digital rectal exam (DRE) findings and with serum tPSA levels ranging from 4 to 10 ng/mL. A report based on a multi-center study on 892 men showed that phi values significantly enhanced the specificity for prostate cancer detection in men with PSA in the 4 to 10 ng/mL PSA range. A score of 27.0 showed 90% sensitivity and 31.1% specificity for predicting prostate cancer at biopsy [28]. A meta-analysis of 16 published studies [29], assessing phi for the detection of high-grade (Gleason Score ≥ 7) prostate cancer, reported a pooled sensitivity of 0.90 and pooled specificity of 0.17 (Area under curve, AUC 0.67 for high-grade disease). More recently, a multi-center study, based on 658 men with PSA levels of 4 to 10 ng/mL, showed that phi outperformed free and total PSA for prostate cancer detection and improved the prediction of high-grade, clinically-significant prostate cancer. The phi had a higher AUC (0.698) compared with %fPSA (0.654), p2PSA (0.550), and PSA (0.549) for clinically-significant prostate cancer based on the Epstein criteria [30]. Many studies, including numerous international multi-center studies, have indicated that the phi score outperforms its individual components for the prediction of overall and high-grade prostate cancer [28,29]. Lazzeri et al. also showed that p2PSA or phi significantly improved the prediction of biopsy outcome over total and free PSA, in a large PRO-PSA Multicentric European Study based on 646 European men from five centers undergoing prostate biopsy for a PSA of 2 to 10 ng/mL or suspicious DRE [31]. The 2015 National Comprehensive Cancer Network (NCCN) guidelines recommended the use of phi for early prostate cancer detection, but the panel did not recommend its use as first-line screening of all patients because of “limited prospective analyses in US populations”. Nonetheless, the panel states that a phi score greater than 35 provides an estimate of the probability of prostate cancer and is “potentially informative in patients who have never undergone a biopsy or had a negative biopsy” [32].

### 2.3. 4Kscore

The 4Kscore^®^ Test (OPKO Lab, Nashville, TN, USA) provides a score that represents a combination of the four kallikrein proteins: tPSA, fPSA, intact PSA, and hK2. The 4K Score test has been included in the NCCN Prostate Cancer Early Detection guidelines since 2015 [33]. The 4Kscore, along with clinical information (such as age and history of prior negative biopsy) provides an estimate of the probability of biopsy positive prostate cancer. Vickers et al. [34] reported that the 4Kscore enhanced the predictive accuracy for clinically diagnosed prostate cancer, compared to total PSA and age. Additionally, 4Kscore provided good accuracy in predicting aggressive disease [35]. A recent meta-analysis of studies evaluating the 4Kscore showed a statistically significant improvement of 8% to 10% in the predictive accuracy of prostate cancer on biopsy and about 48% to 56% of un-necessary biopsies could be avoided [36]. It has also been shown to predict risk for distant prostate cancer metastases, occurring up to 20 years later, from a blood sample in healthy men who have PSA ≥ 2 ng/mL. A study based on 12,542 men from Sweden revealed that among men with PSA > 2 ng/mL at age 50, AUC for distant metastasis was 0.75 for total PSA alone compared with 0.86 for the 4k model; for men with PSA > 2 ng/mL at age 60, AUC increased from 0.805 to 0.875. In 50 and 60-year-old men with an elevated PSA, a low risk 4Kscore predicted low risk for metastasis. Sixty-year-old men with a PSA ≥ 3 ng/mL and a 4Kscore < 7.5% had a <1% chance of developing metastatic prostate cancer by year 15 [37]. The 4Kscore has yet to be FDA approved.

### 2.4. Circulating Tumor Cell (CTC) and Circulating Cell Free (cf) or Tumor (ct) DNA

The CTCs have been actively investigated as biomarkers in patients with advanced prostate cancer. Numerous studies have reported that tumor cells can be detected in the blood and bone marrow of prostate cancer patients [38,39,40,41,42,43,44,45]. With the advent of next generation sequencing (NGS) and sensitive CTC detection assays performed in the blood-plasma of cancer patients, CTCs and circulating cell free tumor DNAs (ct-DNA) are emerging as promising liquid biopsy tools in oncology. These assays offer minimally-invasive approaches of monitoring tumor burden, as well as tumor genome/biology in pre- and post-treatment scenarios. The CellSearchTM (Janssen Diagnostics, Raritan, NJ, USA) is a FDA-approved platform for prognostic use in prostate cancers, which is based on detection and enumeration of CTCs using immunomagnetic capture and fluorescence imaging technology. The CellSearchTM CTC test is an independent predictor of overall and progression-free survival in metastatic prostate cancer [46]. A study by de Bono et al. [46], in a cohort of 276 patients, found that out of 231 evaluable patients, patients with unfavorable pretreatment CTC (>5 CTC/7.5 mL blood) had shorter overall survival (OS) (median OS, 11.5 vs. 21.7 months; Cox hazard ratio, 3.3; *p* < 0.0001). Unfavorable posttreatment CTC counts also predicted shorter OS at 2 to 5, 6 to 8, 9 to 12, and 13 to 20 weeks (median OS, 6.7–9.5 vs. 19.6–20.7 months; Cox hazard ratio, 3.6–6.5; *p* < 0.0001). CTC counts predicted OS better than PSA decrement algorithms at all time points; area under the receiver operator curve for CTC was 81% to 87% compared to 58% to 68% for a 30% PSA reduction. Current CTC research focuses on molecular characterization of CTCs to discover biomarkers for the prediction of treatment response. Molecular analysis from CTCs is found to be comparable to primary tumor tissue and/or metastasis [47] and hence, provides a real-time overview of metastatic prostate cancer [48,49]. The diagnostic and therapeutic potential of CTCs have been explored by numerous studies, revealing that CTCs are promising prognostic and predictive biomarkers for clinical outcome and treatment response in prostate cancer [50,51].

An alternative to CTCs is ct-DNAs or cf-DNAs, which are small nucleic acid fragments that are released from tumor cells. A recent study on 1005 patients with nonmetastatic, clinically detected cancers found that the CancerSEEK test, a blood test based on mutations in cf DNA and levels of circulating proteins in blood, was positive in a median of 70% of the eight cancer types (ovary, liver, stomach, pancreas, esophagus, colorectum, lung, or breast). Test sensitivity ranged from 69% to 98% for five cancer types (ovary, liver, stomach, pancreas, and esophagus) for which there are no screening tests available for average-risk individuals. The specificity of CancerSEEK was greater than 99% [52]. Pantel and colleagues reported that ct-DNA in the blood-plasma of prostate cancer patients may be used as a biomarker [53]. Azad et al. described *AR* gene aberrations, such as AR amplification, in ct-DNAs of castration resistant prostate cancer (CRPC) [54]. Wyatt et al. [55] showed that genomic alterations in cell free (cf) DNA, including *AR* amplification, *AR* mutation, and *RB1* loss, strongly associated with enzalutamide resistance in CRPC patients. While emerging technologies have clear advantages over existing ones, these also have limitations. Some of the challenges include an extremely low abundance of ct-DNA or CTC. These uncertainties can lead to over-diagnosis/treatment as well as psychological and unnecessary financial burdens. Additionally, the correlation of CTC/ct-DNA with the age/sex of patients is also not well established. While liquid biopsy has already found its niche in oncology, especially during treatment of metastatic breast and lung cancers, its utility in other cancers, including CRPC, needs to be extensively evaluated.

### 2.5. Serum Protein Panel

A recent study evaluating 500 serum specimens reported the development of a novel serum protein panel of three prostate cancer biomarkers, Filamin A, Filamin B, and Keratin-19 (FLNA, FLNB, and KRT19) using multivariate models for disease screening and prognosis [56]. The combination of these prostate biomarkers with PSA testing was better than PSA alone in identifying prostate cancer (AUC for panel of FLNA, FLNB, age, PSA, 0.64; PSA alone AUC, 0.58) and improved the prediction of high risk disease (AUC for panel of FLNB, age, and PSA, 0.81; PSA alone AUC, 0.71), low risk disease (AUC for panel of FLNB, age, PSA, and low Gleason Score, 0.72; PSA alone AUC, 0.63), and the prediction of cancer versus benign prostatic hyperplasia (AUC for panel of FLNA, KRT19, and age with PSA, 0.70; PSA alone AUC, 0.58).

## 3. Urine-Based Biomarkers

### 3.1. Prostate Cancer Antigen (PCA3)

Prostate cancer antigen 3 (*PCA3*) was discovered through transcriptome evaluations of normal and tumor prostate tissues [57]. The first reported prostate tissue-specific and prostate cancer-associated, long non-coding (lnc) RNAs were differential display 3 (*DD3*)/(*PCA3*) and a prostate cancer gene expression marker 1 (*PCGEM1*) [58]; however, *PCA3* was found to be over-expressed in virtually all prostate cancers [59,60,61].

The diagnostic utility of *PCA3* RNA in post-DRE urine has been extensively evaluated [62,63,64,65,66]. The Progensa PCA3 assay (Progensa Test Kit, Hologic, Marlborough, MA, USA) is an FDA-approved prostate cancer diagnostic test for use in men aged 50 years or older with elevated serum PSA and previous negative biopsy results [67]. The Progensa PCA3 assay has been included in the European Association of Urology (EAU) guidelines for repeat biopsy decision making. *PCA3* scores reflect the ratio of *PCA3* RNA molecules to *PSA* RNA molecules detected in a patient’s urinary specimen following a DRE. *PCA3* has been shown to have variable sensitivity, specificity, positive predictive value (PPV), and negative predictive value (NPV), depending on the cut-off score chosen (*PCA3* score of 25 or 35). A *PCA3* score <25 is associated with a decreased chance of prostate cancer on subsequent repeat biopsy. A *PCA3* score of 35 was associated with a sensitivity of 58% to 82%, specificity of 58% to 76%, PPV of 67% to 69%, NPV of 87%, and an AUC of 0.68 to 0.87 (95% CI for 0.87: 0.81–0.92) [68,69,70]). Based on several studies on large patient cohorts, urine *PCA3* is found to be a superior serum PSA for prostate cancer diagnosis [71,72,73,74,75]. Whitman et al. also found that urinary *PCA3* before RP could predict extracapsular extension and tumor volume; however, the predictive value of *PCA3* for aggressive disease has been contradictory [66,71].

### 3.2. Multiplex Biomarker Analysis

#### 3.2.1. *TMPRSS2-ERG Fusion* and *PCA3*

The genomic rearrangements leading to the fusion of the AR-regulated *TMPRSS2* gene promoter and the N-terminally deleted *ERG* coding sequence represents the most common prostate cancer-specific driver gene alteration [60,76,77]. The oncogenic activation of *ERG* by these gene fusions is detected in 50% to 65% of prostate cancer patients of European ancestry [60,78]. A multi-center study of *TMPRSS2:ERG* (*T2-ERG*) and *PCA3* in post-DRE urine from 1312 men showed significant improvement over serum PSA for the detection of clinically-relevant cancer at biopsy [79]. In a test developed by the University of Michigan, MLabs for post-DRE urine called the Mi-Prostate Score, *TMPRSS2-ERG* was shown to have a low sensitivity of 24.3% to 37%, but a specificity of 93%, with a PPV of 94% [63,68,76,80]. Further, in combination with serum PSA (cut-off of 10 ng/mL) and urinary *PCA3*, *T2-ERG* provides an improved accuracy in diagnosing prostate cancer with a sensitivity and specificity of 80% and 90%, respectively [73]. This test also provides a risk assessment for aggressive disease [81]. In one study, *TMPRSS2-ERG* gene fusions were highly associated with a Gleason score of ≥7 and prostate cancer-related death [82]. *PCA3* and *TMPRSS2-ERG* in combination with the Prostate Cancer Prevention Trial risk calculator may help in deciding the urgency of biopsy after an elevated serum PSA [81,83]. Rice et al., who analyzed ERG mRNA in post-DRE urine from 237 men, also reported a predictive accuracy of 0.80 for prostate cancer diagnosis in CA men with a PSA ≤ 4.0 ng/mL [79]. Chen et al. also reported that *ERG* and *PCA3* scores had a significantly higher AUC as compared to serum PSA alone (0.68 vs. 0.53, *p* = 0.0062) for predicting prostate cancer in CA patients. Recently, three studies have further underscored the diagnostic utility of *TMPRSS2:ERG* and *PCA3* in urine [68,84,85,86]. Taken together, data for these studies suggest that a combination of urine *PCA3* and *T2-ERG* along with serum PSA significantly improved the detection of aggressive disease (Gleason score ≥ 7) on initial biopsy and a reduction of 42% of unnecessary biopsies.

#### 3.2.2. SelectMDx (DLX1, HOXC6)

The SelectMDx (MDxHealth, Irvine, CA, USA) assay provides the likelihood of detecting prostate cancer upon biopsy, and the probability for high-grade versus low-grade disease, with an area under the curve (AUC) of 0.89 (95% CI: 0.86–0.92). SelectMDx is a qRT-PCR assay performed on post-DRE urine specimens from patients with clinical risk factors for prostate cancer, who are being considered for biopsy. The test measures the mRNA levels of the *DLX1* and *HOXC6* genes, using *KLK3* expression as an internal reference, to aid in patient selection for prostate biopsy. A three-gene panel consisting of *HOXC6*, *TDRD1*, and *DLX1* [87] was also found to significantly improve the detection of clinically-significant prostate cancer with an AUC of 0.77 when compared to PSA (AUC 0.72) and *PCA3* (AUC 0.68) Van Neste et al. recently validated and compared these markers with other urine-based prostate cancer gene markers [88].

#### 3.2.3. ExoDx Prostate (IntelliScore) (EPI)

ExoDx prostate Intelliscore (Intelliscore) (Exosome Diagnostics Inc., Cambridge, MA, USA) is a regular urine (non-DRE) exosome-based assay that measures *ERG* and *PCA3* along with *SPDEF* as an internal reference. The test is indicated for men age aged ≥50 years with a PSA of 2 to 10 ng/mL being considered for initial prostate biopsy. It is used in conjunction with the current standard of care (SOC) variables (i.e., PSA level, age, race, and family history) to determine the risk of Gleason 6, Gleason 7, and benign disease on initial biopsy [89,90]. An evaluation of the ExoDx score (EXO106) in 195 patients at the diagnostic biopsy stage showed that the EXO106 score demonstrated good clinical performance in predicting biopsy results for both any cancer and high-grade disease. For high grade disease, it significantly (*p* = 0.0009) improved the predictive performance of SOC (EXO106 and SOC, AUC 0.80; SOC alone, AUC 0.67). A multicentric study, based on 774 patients (training cohort, 255 patients; validation cohort, 519 patients) from 22 community practice and academic urologic sites in the US, found that the ExoDx assay plus SOC features were significantly better at predicting the presence of Gleason grade ≥ 7 cancer and negative biopsy results than the ExoDx assay or SOC data alone (training cohort, AUC 0.77; validation cohort, AUC 0.73) [89]. For high-grade prostate cancer, using a predefined cut point, 27% of biopsies would have been avoided, missing only 5% of patients with dominant pattern 4 high-risk GS7 disease [89]. The EPI test was further validated in phase I of the two phase prospective adaptive decision impact trial on 500 patients. It confirmed the accuracy of the EPI test for discriminating high grade prostate cancer of ≥GG2 from benign and GG1 biopsies in men aged ≥50 years with a PSA of 2 to 10 ng/mL [91].

## 4. Tissue-Based Biomarkers

### 4.1. ConfirmMDx

ConfirmMDx (MDxHealth, Inc, Irvine, CA, USA) is a prostate tissue biopsy-based, DNA methylation assay. This multiplexed, epigenetic test evaluates Glutathione S-Transferase Pi 1 (*GSTP1*), Adenomatous Polyposis Coli (*APC*), and Ras association (*RalGDS*/*AF-6*) domain family member 2 (*RASSF2*). Complementing the *GSTP1*, methylation of *APC* and *RASSF2* genes is also frequently found in prostate cancer. Intriguingly, these markers have demonstrated a useful “field effect,” meaning a positive ConfirmMDx test for pathology determined cancer negative biopsy suggests a cancer cells/region was missed by the biopsy procedure. It is commercially available and aims to predict true negative prostate biopsies from those with possible occult cancer [92]. The test is included in National Comprehensive Cancer Network (NCCN) guidelines for men with at least one prior negative biopsy [29]. It has been validated by two studies: The Methylation Analysis to Locate Occult Cancer (MATLOC) trial, where the assay was significantly associated with patient outcome with an odds ratio of 3.17 [93], and the Detection Of Cancer Using Methylated Events in Negative Tissue (DOCUMENT)study, where the assay was independently associated with prostate cancer detection in a repeat biopsy with an 88% negative predictive value [94]. The clinical utility of the ConfirmMDx assay in US urologic practices is being evaluated by the ongoing Prostate Assay Specific Clinical Utility at Launch (PASCUAL) study, which is expected to be completed in 2018 (ClinicalTrials.gov identifier, NCT number: NCT02250313).

### 4.2. Oncotype DX Genomic Prostate Score (GPS) 

The Oncotype DX GPS test (Genomic Health, Redwood City, CA, USA) is a biopsy-based genomic test, which measures mRNA expression of 17 genes responsible for the growth and survival of tumor cells. The assay was developed and studied in 4500 patients [95]. The development phase of this test used reverse transcription-polymerase chain reaction (RT-PCR)-based assays and formalin fixed paraffin embedded (FFPE) radical prostatectomy (RP) specimens from 441 low to intermediate risk prostate cancer patients. Of the 732 selected genes, 288 genes predicted clinical recurrence (local recurrence or distant metastasis) and 198 genes were associated with aggressive disease after adjustment for PSA, Gleason score (GS), and clinical stage [96]. This was followed by definition of a 17 gene prognostic expression signature as GPS. In a validation study, based on 395 needle biopsies from patients under active surveillance (AS), GPS predicted high-grade (odds ratio (OR) per 20 GPS units: 2.3; 95% confidence interval (CI), 1.5–3.7; *p* < 0.001) and high-stage (OR per 20 GPS units: 1.9; 95% CI, 1.3–3.0; *p* = 0.003) at surgical pathology. Additionally, GPS predicted high-grade and/or high-stage disease, after controlling for established clinical factors (age, PSA, clinical stage, biopsy GS), resulting in an OR of 2.1 (95% CI, 1.4–3.2) when adjusting for the Cancer of the Prostate Risk Assessment score (CAPRA). The Oncotype DX GPS assay was independently validated in a biopsy cohort of 431 racially diverse men (with very low-, low-, or intermediate-risk prostate cancer) treated at the Department of Defense medical centers [97]. These results confirmed that the GPS strongly correlated with BCR, metastasis, and adverse pathology in RP specimens. Of note, this study underscored the observation that the GPS assay performed similarly in AA and CA patients. Taken together, these studies continue to highlight the prognostic value of the Oncotype DX GPS test in the management of low and intermediate risk patients.

### 4.3. Decipher

The Decipher test (GenomeDx, San Diego, CA, USA) is also a genomics-driven test, which measures the RNA expression levels of 22 different genes. The genes were selected based on unique patterns of differential expression for 192 early metastatic cases (within 5 years of rising PSA) in comparison to 271 controls in a retrospective, nested case-control study [98]. These genes are involved in important biological pathways, including cell cycle progression, cell proliferation, differentiation, adhesion, AR and immune response regulating signallingpathways. The 22-marker signature, known as the Genomic Classifier (GC), is available for both RP and prostate biopsy specimens and is depicted as a score from 0 to 1.0 (a higher score indicates a higher probability of clinical metastasis). According to NCCN guidelines, it is recommended for patients with adverse pathology after RP [99]. The Decipher test calculates the probability of clinical metastasis within 5 years of RP and 10-year prostate cancer-specific mortality in men with high risk pathology or high-risk clinical features after RP [98,100,101,102]. The test showed a very high discrimination in predicting clinical metastasis (AUC, 0.75–0.83) and cancer-specific mortality (AUC 0.78) in validation studies, which significantly outperformed available clinicopathologic characteristics (AUC 0.69) [98,101,102,103]. Recently, the Decipher biopsy test has been validated and provides risk at RP for pathologic grade upgrading (Gleason pattern 4 or 5), 5-year development of metastasis, and 10-year PCSM [104]. Further, the timing of postoperative radiotherapy (adjuvant vs. salvage) may be guided based on Decipher scores. The Decipher test was also the only independent predictor of clinical metastasis in patients with biochemical recurrence after surgery [103].

### 4.4. ProMark

The Promark (Metamark, Cambridge, MA, USA) is a protein-based prognostic assay, which predicts cancer aggressiveness in patients with biopsy Gleason scores of 3 + 3 and 3 + 4. The test evaluates eight protein markers using an automated, quantitative, and multiplex immunofluorescence assay on FFPE tissues [105]. The protein marker panel provides a score of 0 to 1, which predicts adverse pathology at time of RP for patients who are NCCN very low-risk or low-risk considering active surveillance. In an initial study [106], the risk score for favorable and unfavorable pathology was defined in 381 patient biopsies with matched prostatectomy pathology. At a risk score ≤ 0.33, the predictive values for favorable pathology in very low-risk and low-risk NCCN groups and low-risk D’Amico groups were 95%, 81.5%, and 87.2%, respectively, which was higher than the current risk classification groups themselves (80.3%, 63.8%, and 70.6%, respectively). At a risk score > 0.8, the predictive value for unfavorable pathology was 76.9% across all risk groups.

## 5. Promising Studies

### 5.1. Exosomal Biomarkers

Exosomes are small (30–150 nm) double lipid membrane bound extracellular vesicles containing proteins, lipids, and nucleic acid and are secreted by most cell types. Biomarker discovery using exosomes is gaining significant interest considering that they can be detected and are remarkably stable in body fluids, including blood, urine, semen, saliva, and cell culture medium [107,108]. Prostate derived exosomes are a rich source of molecular markers (protein, RNA, miRNA) for prostate cancer diagnosis and prognosis [109,110]. Prostasomes or prostate derived exosomes are found to be higher in prostate cancer patients and were correlated with high Gleason scores [111]. Urine exosome derived *PCA3* and *TMPRSS2-ERG* have been found to associated with prostate cancer diagnosis [112] and prostate cancer risk stratification [89,91]. Changes in miR concentration can be detected in plasma and serum samples, which can be useful as an aid in the diagnosis of prostate cancer [113,114,115]. miR-141 and miR-375 from serum derived exosomes was found to be associated with metastatic prostate cancer [114]. miR 34a was also found to be a predictive marker for the response to docetaxel [116]. Like mRNA or miRNA, exosomal proteins are also found to be useful prostate cancer biomarkers. Plasma derived exosomal survivin and claudin 3 was also found to be high in prostate cancer patients, suggestive of its role in prostate cancer diagnosis [117,118,119].

### 5.2. Tumor Micro-Environment (TME) Associated Biomarkers

Tumorigenesis is highly associated with the interactions of cancer cells with their tumor microenvironment (TME). The interaction between the extracellular matrix (ECM) and stromal cells may further determine if the primary tumor is eradicated, metastasizes, or leads to dormant micrometastases [120,121]. Important components of the TME are the ECM, fibroblasts and myofibroblasts, mesenchymal stem cells, neuroendocrine cells (NE), adipose cells, immune and inflammatory cells, and the blood and lymphatic vascular networks. Galectin-1 overexpression in cancer associated fibroblasts (CAFs) is correlated with poor prognosis in several types of cancer, including prostate cancer [122]. In addition, Jung et al. reported that Chemokine (C–X–C motif) ligand 12 (CXCL12) in CAFs was found to induce epithelial-mesenchymal transition (EMT) in prostate cancer [123]. On the other hand, NE cells played a key role in the development of NE cell-tumor by influencing the proliferation and aggressiveness of prostate cancer cells [124]. Immune cells, such as Tregs, Th17 cells, and macrophages, are also involved in prostate cancer progression while cytokines, such as IL-6 and RANKL, secreted by cells in the TME have been found to have a pleiotropic effect on prostate cancer cells [125]. Clinical studies have established the differential role of AR or AR signaling pathways in target tissues depending on cell type, TME, and hormone levels. Stromal AR has also been shown to mediate prostate cancer metastasis [126]. Reciprocal AR responses in the epithelium and stroma regulate prostate development and homeostasis while aberrant responses might result in tumorigenesis [127]. These findings show that the TME offers potentially promising targets for prostate cancer therapy.

## 6. Developmental Studies

### 6.1. Prostate Core Mitomic Test (PCMT)

The Prostate Core Mitomic Test (MDNA Life Sciences Inc.) is a proprietary tissue-based test based on evidence linking mitochondrial function with regulation by oncogenes and tumor suppressors [128]. The goal of the test is to correctly identify true negative prostate biopsies [129]. In a clinical validation study, PCMT was associated with a sensitivity of 84%, specificity of 54%, and a negative predictive value of 91% [130].

### 6.2. Prostarix Risk Score

Prostarix Risk Score (Boswick Laboratories) is a post-DRE urine-based test, which aims to help physicians decide if an initial or repeat biopsy is necessary for patients with a negative DRE and mildly elevated PSA levels [131]. The method quantitatively measures the 4-metabolite panel (sarcosine, alanine, glycine, and glutamate) in post-DRE-derived urine. The performance of the metabolite panel (AUC = 0.64) was significantly greater than either PSA (AUC = 0.53) or the PCPT risk calculator (AUC = 0.61) alone [132].

### 6.3. Urine PCA3, PSGR, and PSMA

Prostate-specific G-protein coupled receptor (*PSGR*) has been reported to be over-expressed in prostate cancer [133]. Using post-DRE urine specimens, the combined analysis of *PSGR* and *PCA3* was performed by qRT-PCR. The specificity for prostate cancer detection on biopsy could be improved by fixing the sensitivity at 95%, with individual specificities of 15% for *PSGR*, 17% for *PCA3*, and 34% for the *PSGR* and *PCA3* duplex panel [134]. Furthermore, by fixing the sensitivity at 96%, the combined detection of *PSMA*, *PSGR*, and *PCA3* increased the specificity to 50% [135].

### 6.4. Detection of Prostate Cancer Cells in Urine

There are reports of assay development to detect intact prostate cancer cells in post-DRE urine. Using post-DRE urine, prostate cancer cells have been detected using multiple color fluorescent staining for AMACR, Nkx3.1, Nucleolin, and DAPI. The multiplex immunofluorescence cytology test achieved a sensitivity of 36% (9/25), and specificity was 100% (8/8) for cancer detection in a cohort of 50 patients [136]. Another study also described a post-DRE urine-based prostate cancer marker panel (UCMP) by developing a method for cell capture on a membrane followed by an evaluation of prostate cancer-associated proteins using immuno-cytochemistry. The protein expression profile included ERG, AMACR (prostate cancer associated), and Prostein (prostate epithelium specific) of cells captured from the post-DRE urine specimens [137]. An assay sensitivity of 64% and a specificity of 68.8% for prostate cancer detection at biopsy was achieved in a cohort of 63 patients.

## 7. Emerging Role of Molecular Biomarkers in the Context of Clinical Management of Prostate Cancer

The emerging, more precise prostate cancer molecular biomarkers hold tremendous potential in improving risk assessment, reducing overtreatment, and providing more selective treatment for patients with high-risk disease. A schematic representation (Figure 1) depicts the use and appropriate timing of some of the well-studied biomarkers in the clinical management of a patient with suspected or proven prostate cancer. The phi, 4Kscore, and PSA tests provide information on which patients should be referred for biopsy. The PSA, 4Kscore, phi, SelectMDx, *PCA3*, and Confirm MDx assays provide information on suspected prostate cancer patients who need to undergo initial biopsy. The assays, including *PCA3*, *TMPRSS2-ERG*, ExoDx Intelliscore, Select MDx, and Confirm MDx, have potential for patients who need to be re-biopsied, where the initial biopsy was found to be negative for cancer. The prognostic biomarker tests, Prolaris and Oncotype DX, provide prostate cancer risk stratification, informing which patients need to be treated after positive biopsy for prostate cancer. Additionally, prognostic markers may also provide information on patients who need to be treated post-surgery (Prolaris, Decipher).

## 8. Conclusions

Emerging prostate cancer molecular biomarkers noted in this review are beginning to play an important role in improving diagnosis and treatment. The good news is that there is already a promising collection of biomarkers; however, there is also a real challenge for clinicians and reimbursing agencies on how to assess and prioritize the new markers. These biomarkers have the potential to help clinicians in determining a need for biopsy, especially in patients with PSA levels in a gray zone of 4.0 to 10.0 ng/mL, thus avoiding un-necessary biopsies. These biomarkers may also play an important role in distinguishing clinically significant prostate cancer, thereby determining the treatment regime. Also in the pipeline are liquid biopsy, multi-omics, advanced imaging, and an integration of these technologies. For example, clinically validated robust biomarkers along with an integrated imaging approach, such as multiparametric magnetic resonance imaging (MRI), may provide a more individualized risk assessment for prostate cancer diagnosis and management of the disease.

Challenges in cancer biomarker development, including prostate cancer, require extensive validation and cross validation, head to head comparisons of the potential biomarkers, development of marker panels applicable to multiple clinical contexts and therapies, affordability, and access. Potential biomarkers should provide additional independent information from already established clinical and pathological variables, to improve the predictive accuracy for prostate cancer diagnosis, prognosis, and treatment response. The available and expanding body of literature and pace in prostate cancer biomarker research, together with the development of highly integrated supercomputing platforms, are likely to lead to more exciting discoveries that will enhance individualized risk assessment and clinical management of the disease. The multi-center studies supported by biotech/pharma companies, as well as government agencies, such as the National Cancer Institute (NCI)-Early Detection Research Network (EDRN), will continue to provide much needed guidelines and resources toward these developments.

## Figures and Tables

**Figure 1 ijms-20-01813-f001:**
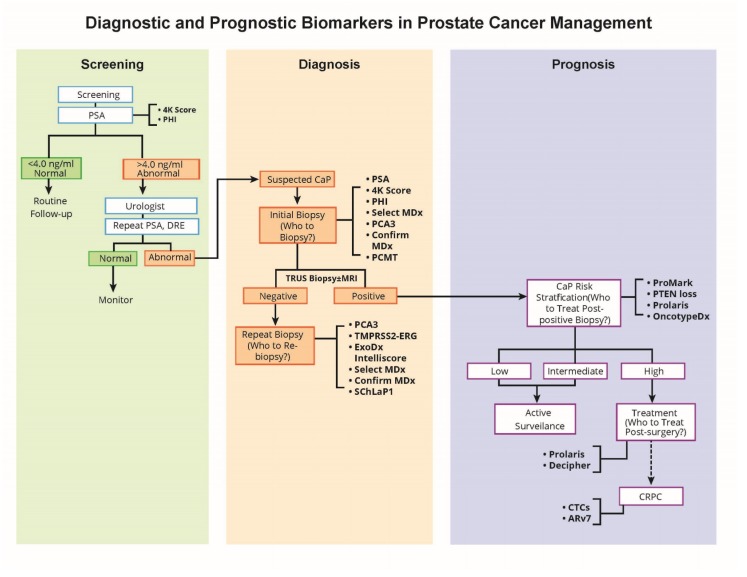
Role of biomarkers in prostate cancer management.

**Table 1 ijms-20-01813-t001:** Current Food and Drug Administration (FDA) or Clinical Laboratory Improvement Amendments (CLIA) approved blood-, urine-, and tissue-based biomarkers in prostate cancer.

Biomarker Test	Molecular Markers	Available as
**Serum-based**		
Prostate Serum Antigen (tPSA)	PSA	FDA
PHI (Beckman Coulter Inc., Brea, CA, USA)	Total PSA, fPSA, p2PSA	FDA
4K (OPKO lab, Miami, FL, USA)	Total PSA, fPSA, intact PSA, hK2	CLIA-approved
**Urine-based**		
PCA3 (Progensa) Hologic, Marlborough, MA, USA	*PCA3*	FDA
ExoDX Prostate (Intelliscore) Exosome Diagnostics Inc., Waltham, MA, USA	Exosomal RNA (*PCA3, ERG*)	CLIA-approved
MiPS (Detroit, MI, USA)	*PCA3* and *TMPRSS2-ERG* mRNA	CLIA-approved
SelectMDX (MDx Health, Irvine, CA, USA)	*HOXC6, DLX1*	CLIA-approved
**Tissue-based**		
ConfirmMDx (MDxHealth, Irvine, CA, USA)	DNA hypermethylation (*GSTP1; APC; RASSF1*)	CLIA-approved
Prolaris (Myriad Genetics, Salt Lake City, UT, USA)	mRNA expression; 31 genes (cell cycle progression)	FDA
Oncotype Dx (Genomic Health, Redwood City, CA, USA)	mRNA expression; 17 genes	CLIA-approved
Decipher (GenomeDx Biosciences, San Diego, CA, USA)	mRNA expression; 22 genes (cell proliferation, migration, tumor motility, androgen signaling, and immune system evasion)	CLIA-approved
Promark (Metamark, Cambridge, MA, USA)	Protein biomarker test (8 proteins)	CLIA-approved

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
