# Peer review of "A Rich Array of Prostate Cancer Molecular Biomarkers: Opportunities and Challenges"

_ijms, 2019, doi:10.3390/ijms20081813_

Round 1
Reviewer 1 Report
This is an intriguing manuscript regarding the state of the art of current prognostic and diagnostic biomarkers in Prostate cancer (PC). In this manuscript, Dr. Kohaar and collegues focus also on biomarkers that are under development discussing their utility in clinical approaches.
Tables and figures are representative and well performed and the english style is also good.
I have two questions:
1) In these last years, exosomes are emerging as useful non-invasive biomarkers for different types of cancer. In literature are present also data about exosomes derived form PC. The authors should consider and add information about this aspect.
2) Tumor microenvironment is also important for cancer progression and it is an emerging field.
The authors should consider for example doi: 10.3389/fendo.2014.00225 and related findings and add information about tumor microenvironment in PC.
From points 1 and 2 the authors should explain if there are new biomarkers.
Author Response
1) In these last years, exosomes are emerging as useful non-invasive biomarkers for different types of cancer. In literature are present also data about exosomes derived form PC. The authors should consider and add information about this aspect.
Response 1: As per reviewer’s suggestion, we have included a section on exosomal markers (section 5.1) on page 9. Studies based on urinary exosomal markers are also covered under section 3.2.3 [(ExoDx Prostate (IntelliScore) (EPI)].
2) Tumor microenvironment is also important for cancer progression and it is an emerging field.
Response 2: As per reviewer’s suggestion, we have included a section on tumor micro-environment associated biomarkers (Section 5.2) on page 9. Studies based on circulating tumor cell (CTC) and circulating cell cree (cf) or tumor (ct) DNA are also covered under under blood-based biomarkers (section 2.4) on pages 4 and 5.
As suggested by the reviewer, we have included information on recent studies in both sections [section 5.1 (Exosomal markers) and section 5.2 (Tumor micro-environment associated Biomarkers)].
This article provides an in depth review of commercially available prostate cancer diagnostic and prognostic tests. Emphasis is placed on explaining why better tests are needed and when testing should be performed/how test data can be used. The scientific basis for each test, test details, and test validity are thoroughly described. This is an excellent article, however, I do have some concerns/suggestions and these are listed below;
1. There are several examples of awkward phrasing in the Abstract, however, the rest of the article is well written. I suggest that the authors consider editing the Abstract to make it easier to read and follow.
2. Lines 14 – 16: I think this sentence needs to be rewritten to further emphasize the importance of being able to distinguish between indolent and aggressive prostate cancers. This is explained in the Introduction section but needs to be touched on in the Abstract too. Perhaps make note of survival rates for patients who experience biochemical recurrence/develop castration resistant prostate cancer to help emphasize the need for the development of prognostic tests.
3. Line 18: I suggest replacing the term ‘biomarkers’ with ‘diagnostic and prognostic tests’ (there have been multiple biomarkers identified but not all of them have been developed into commercially available tests)
4. Line 28: a citation needs to be added for the first sentence
5. Lines 30: please use more up to date statistics (states ‘2012’, more recent data are available)
6. Line 45: I think it would be helpful to include definitions of specificity and sensitivity here (simply put a brief description in brackets next to these terms).
7. Lines 71 – 73: The authors should discuss potential reasons why AA men presented with higher PSA/higher tumor volumes compared to their Caucasian counterparts, and comment on current disparities in testing/how they are being addressed.
8. Lines 83 – 85: I think it would be helpful to note that the PSA test has much higher accuracy when used for surveillance after diagnosis or after treatment compared to when used for initial diagnosis purposes
9. Lines 131- 133: The authors should make note of the ability of this test to predict which patients will experience metastatic prostate cancer (in addition to stating its ability to predict which patients are unlikely to develop metastatic disease)
10. Lines 142 – 145: The authors should note details of test performance for the CellSearch assay. These details are provided for other assays.
11. Section 3.3: See comment ‘10’; please include test performance details
12. I suggest moving the urine-based biomarkers section to after the blood-based biomarkers section because there are many parallels between these (the authors may wish to comment on parallels and differences/challenges associated with urine versus blood-based tests, e.g. isolation of exosomes)
13. Figure 1: The text in this figure is very small making it difficult to read. Please consider increasing the font size/reformatting this figure
Author Response
Response to Reviewer 2 Comments
Point 1: There are several examples of awkward phrasing in the Abstract, however, the rest of the article is well written. I suggest that the authors consider editing the Abstract to make it easier to read and follow.
Response 1: We have made the necessary suggested change and thank the reviewer.
Point 2: Lines 14 – 16: I think this sentence needs to be rewritten to further emphasize the importance of being able to distinguish between indolent and aggressive prostate cancers. This is explained in the Introduction section but needs to be touched on in the Abstract too. Perhaps make note of survival rates for patients who experience biochemical recurrence/develop castration resistant prostate cancer to help emphasize the need for the development of prognostic tests.
Response 2: We have made the necessary suggested change (lines 14-17).
Point 3: Line 18: I suggest replacing the term ‘biomarkers’ with ‘diagnostic and prognostic tests’ (there have been multiple biomarkers identified but not all of them have been developed into commercially available tests)
Response 3: As suggested the ‘biomarkers’ in line 19 is replaced with ‘diagnostic and prognostic tests.’
Point 4: Line 28: a citation needs to be added for the first sentence
Response 4: As per reviewer’s suggestion, citation is added for the first sentence (line 29).
Point 5: Lines 30: please use more up to date statistics (states ‘2012’, more recent data are available)
Response 5: As per reviewer’s suggestion, 2019 citation is included (line31).
Point 6: Line 45: I think it would be helpful to include definitions of specificity and sensitivity here (simply put a brief description in brackets next to these terms).
Response 6: As per reviewer’s suggestion, we have included the definitions of specificity and sensitivity (lines 45-47).
Point 7: Lines 71 – 73: The authors should discuss potential reasons why AA men presented with higher PSA/higher tumor volumes compared to their Caucasian counterparts, and comment on current disparities in testing/how they are being addressed.
Response 7: As suggested by reviewer, we have included the required information in lines 75-79.
Point 8: Lines 83 – 85: I think it would be helpful to note that the PSA test has much higher accuracy when used for surveillance after diagnosis or after treatment compared to when used for initial diagnosis purposes
Response 8: As suggested by reviewer, a note has been added for higher accuracy of PSA test in active surveillance patients and after treatment (lines 88-90).
Point 9: Lines 131- 133: The authors should make note of the ability of this test to predict which patients will experience metastatic prostate cancer (in addition to stating its ability to predict which patients are unlikely to develop metastatic disease)
Response 9: As suggested by reviewer, we have included the required information in lines 135-141.
Point 10: Lines 142 – 145: The authors should note details of test performance for the CellSearch assay. These details are provided for other assays.
Response 10: As suggested by reviewer, we have included the required information in lines 154-161.
Point 11: Section 3.3: See comment ‘10’; please include test performance details
Response 11: Section 3 has been updated with test performance details (Lines323-330).
Point 12: I suggest moving the urine-based biomarkers section to after the blood-based biomarkers section because there are many parallels between these (the authors may wish to comment on parallels and differences/challenges associated with urine versus blood-based tests, e.g. isolation of exosomes)
Response 12: As suggested, the urine-based biomarker has been moved after blood-based biomarkers (page 5). The present review is based on the current knowledge of available diagnostic and prognostic molecular markers in CaP and discusses their utility in the clinic. We have tried to explain the test performance and utility of biomarkers in each section; however, did not intend to provide comparison of different biomarker platforms, as it will extend the manuscript beyond the scope of the required article. However, we are very thankful to the reviewer for raising this question because there is very limited data on head-to-head comparison of different biomarker platforms including methodological comparisons.
Point 13: Figure 1: The text in this figure is very small making it difficult to read. Please consider increasing the font size/reformatting this figure
Response 13: As suggested by the reviewer, figure has been updated and reformatted for font size and visual clarity.